# Inoculation with Mycorrhizal Fungi and Irrigation Management Shape the Bacterial and Fungal Communities and Networks in Vineyard Soils

**DOI:** 10.3390/microorganisms9061273

**Published:** 2021-06-11

**Authors:** Nazareth Torres, Runze Yu, S. Kaan Kurtural

**Affiliations:** Department of Viticulture and Enology, University of California Davis, 1 Shields Avenue, Davis, CA 95616, USA; nazareth.torres@unavarra.es (N.T.); crzyu@ucdavis.edu (R.Y.)

**Keywords:** arbuscular mycorrhizal fungi, co-occurrence networks, grapevine, microbiome, soil health, water deficit

## Abstract

Vineyard-living microbiota affect grapevine health and adaptation to changing environments and determine the biological quality of soils that strongly influence wine quality. However, their abundance and interactions may be affected by vineyard management. The present study was conducted to assess whether the vineyard soil microbiome was altered by the use of biostimulants (arbuscular mycorrhizal fungi (AMF) inoculation vs. non-inoculated) and/or irrigation management (fully irrigated vs. half irrigated). Bacterial and fungal communities in vineyard soils were shaped by both time course and soil management (i.e., the use of biostimulants and irrigation). Regarding alpha diversity, fungal communities were more responsive to treatments, whereas changes in beta diversity were mainly recorded in the bacterial communities. Edaphic factors rarely influence bacterial and fungal communities. Microbial network analyses suggested that the bacterial associations were weaker than the fungal ones under half irrigation and that the inoculation with AMF led to the increase in positive associations between vineyard-soil-living microbes. Altogether, the results highlight the need for more studies on the effect of management practices, especially the addition of AMF on cropping systems, to fully understand the factors that drive their variability, strengthen beneficial microbial networks, and achieve better soil quality, which will improve crop performance.

## 1. Introduction

Plant microbiomes play an important role in supporting plant health and adaptation to changing environments [1]. The biological quality of soils may be defined as the capacity of a soil to host a large quantity and diversity of living organisms that are involved in its functioning and the provision of ecosystem services [2]. Within these services, the role of microorganisms on decomposition, mineralization of plant nutrients, atmospheric N fixation, and C sequestration is particularly relevant for cropping systems [3]. In vineyards, the composition of the soil microbiome has been recently highlighted because it seems to be the major driver in shaping the bacterial and fungal communities associated with grapevine tissues, including berries [4], defining the regional characteristics of the wine [5,6,7]. Thus, the traditional conception of ‘terroir’, defined as the interaction of the vine with its ecosystem [8], requires the addition of the ‘microbial terroir’ concept given that microbial vineyard inhabitants determine grape characteristics and quality [9,10,11,12]. In addition, the need to reconcile the ‘terroir’ and the soil health concepts in the context of wine production was recently highlighted [3].

The effect of different viticulture management practices on grapevine physiology and berry composition is a recurrent topic in viticulture research. However, less is known about how the microbial communities’ associations with plants and soil are affected. In this regard, previous studies reported that soil microbial diversity varied considerably between vineyards under conventional and ecological management, with bacterial communities strongly affected by tillage [13], whereas others demonstrated that land uses influenced the structure and composition of fungal communities and the geography of the affected bacterial communities [14]. 

Within management practices, irrigation is particularly relevant for grapevines due to many viticulture areas relying on irrigation for crop production [15]. Furthermore, the exacerbation of water deficits due to global warming is rapidly expanding irrigation in traditional rain-fed viticulture regions to mitigate negative effects on grapevines [15]. Irrigation practices are known to disturb the soil ecosystem, affecting soil characteristics and soil microbial functioning [16]. In addition, water availability in soils alters microbial communities, with potentially long-term consequences, including ensuing plant community composition and the ability of aboveground and belowground communities to withstand future disturbances [17]. Despite these reports, the biological quality of winegrowing soils and the impact of viticultural practices are still poorly known, and, to the best of our knowledge, information is lacking on how water management may alter soil microbial communities.

Recent research has suggested that plant symbionts strongly affect the plant-associated microbiome [18]; however, little is known about how the “symbiosis cascade effects” proposed by these authors may shape soil microbial communities in crop production systems. Currently, over 400 microbial products are available in the markets as crop fertilizer or crop management tools [19]. The intensive practices of modern agriculture had a detrimental environmental impact on soils, increasing greenhouse gas emissions, nutrient leaching given the intensive fertilizer application, and soil erosion and decreasing biodiversity [2]. Therefore, searching for environmental-friendly management practices is paramount to alleviate such deleterious effects. Previous studies have suggested that biostimulants such as AMF inoculation might be used for enhancing plant resistance to abiotic environmental stresses because of their effectiveness in improving crop productivity and quality under abiotic stresses [20]. However, only a few studies have been conducted to assess their effect on plant-associated microbiota [21,22,23].

We hypothesize that irrigation management and/or AMF inoculation may affect microbial diversity, enhancing plant-growth-promoting bacteria while decreasing pathogen abundances and edaphic soil characteristics. Therefore, a randomized experiment was conducted to address the effect of irrigation management (half irrigation vs. full irrigation) and AMF inoculation (inoculated vs. non-inoculated) on bacterial and fungal communities living in the bulk soil of a newly established vineyard in its first productive year. 

## 2. Materials and Methods

### 2.1. Site Description and Experimental Design

This study was conducted during the growing season of 2020 on a three-year-old vineyard planted with Merlot (clone 181) grafted onto 3309C rootstock in its first productive year. Soil texture and chemical properties were analyzed before grapevine planting in 2017 by Delavalle Laboratory Inc. (Fresno, CA, USA). The soil was mainly classified as silt with low pH (to ca. 5.7) and electrical conductivity (<0.5 dS/m). It contained low levels of boron (<0.1 mg/L), potassium (<150 mg/kg), and nitrogen and phosphorus salts (<2 and <10 mg/kg, respectively) and high levels of iron and copper (>50 and >2 mg/kg, respectively). Grapevines were planted in 2018 at a density of 3 × 2 m (row × vine), oriented east–west, at the Oakville Experimental Research Station (WGS84 coordinates: 38.429°, −122.410°) in California, USA. Natural vegetation was allowed to grow in the alleys and mowed according to the vineyard manager’s discretion, but a no-till system was applied. In all sampling spots, grasses were mainly constituted from species of Poaceae family, Plantago sp., Trifolium sp., Convolvulus sp., and Barbarea vulgaris. The experimental vineyard was drip-irrigated, with emitters spaced every 2 m along the drip line. The experiment was conducted in a split-plot design with 2 × 2 factors (AMF inoculated or non-inoculated vines subjected to two irrigation amounts) in a randomized complete block with 4 repetitions (a total of 16 sampling plots, as indicated in Appendix A). Each experimental replicate consisted of 15 grapevines occupying 90 m^2^/treatment-replicate.

AMF-inoculated vines (inoculated, I) were drenched around their trunk for 50 s with a diluted AMF inoculum at a 10 g/1000 plants rate, following the manufacturer’s recommendations, using a spot sprayer. The commercial Myco Apply Endo Maxx inoculum (Mycorrhizal Applications LLC, OR, USA) consists of a suspendable powder containing 5625 propagules/g of living propagules of *Rhizophagus intraradices* (basionym *Glomus intraradices*), *Funneliformis mosseae* (basionym *Glomus mosseae*), *Glomus aggregatum*, and *Glomus etunicatum*. In order to restore the rhizobacteria and other soil free-living microorganisms accompanying AMF, non-inoculated vines (non-inoculated, NI) received the same amount of a filtrate inoculum obtained by passing diluted mycorrhizal inoculum through a Grade 5 Whatman filter paper with particle retention of 2.5 µm (Whatman 5; GE Healthcare, MA, USA). AMF inoculum and filtrate application occurred at the beginning of the growing season (20 March). Vineyard crop evapotranspiration (ET_c_) was calculated by multiplying the reference evapotranspiration (ET_o_, California Irrigation Management Information System, CIMIS #77) and the crop coefficient (K_c_). Then, half of the I and NI plants were irrigated weekly (starting on 8 May) with the amount of water needed for full expansive growth (fully irrigated, FI). FI irrigation coincided with the amount of water needed to restore 100% of the ET_c_, while the other half of I and NI plants received half of the water amount of the FI vines (half irrigated, HI).

### 2.2. Soil Sampling, DNA Extraction, and Sequencing

Bulk soil samples for microbial analysis were collected before treatment application on the 16 sampling plots (13 March; not treated, NT) and after three months (16 June) from the same plots, according to the established protocol [24]. Soil sampling was conducted at a depth of 15–20 cm and a distance of 30 cm away from the vine trunks, compiling four different cores within the plot of 90 m^2^/treatment-replicate with a sterilized teaspoon. Each sample from a single plot and time point consisted of ca. 10 mL of soil and was made by pooling together the four subsamples from random spots in each plot. Then, soil samples were placed in sterile tubes and homogenized by manual shaking without sieving. Between each sampling point, digging tools were sterilized with 70% ethanol in order to avoid cross-contamination between plots. 

The 32 (16 + 16) soil samples were immediately sent after each collection for molecular analysis to the Biome Makers laboratory in Sacramento, CA, USA. Nucleic acids were extracted directly from samples using WINESEQ extraction methodology [25] that avoids biases, allowing us to detect the maximum possible diversity values. Extraction was performed using different bead-beating cycles based on DNA extraction kits, such as the DNeasy Powerlyzer Powersoil Kit (Qiagen, Hilden, Germany). The V4 region of the 16S rRNA gene was amplified by PCR using the primers 515F (GTGYCAGCMGCCGCGGTAA) and 806R (GGACTACNVGGGTWTCTAAT). Then, libraries were prepared following the two-step PCR protocol from Illumina and sequenced on an Illumina MiSeq using pair-end sequencing (2 × 300 bp). Libraries were prepared by amplifying the 16s rRNA V4 region and the ITS1 region using Biome Makers^®^ custom primers (Patent WO2017096385). In each reaction in the first PCR for 16S, the mix contained 12 μL of AccuPrime™ SuperMix II (Thermo Scientific™, Waltham, MA, USA), 0.5 μL each of forward and reverse primer from a 10 μM stock, 0.5 μL of bovine serum albumin (BSA) to a final concentration of 0.025 mg/mL, 1.5 μL of sterile water, and 5 μL of the template. The reaction mixture was preincubated at 95 °C for 2 min, followed by 33 (16S) and 40 (ITS) cycles of 95 °C for 15 s, 55 °C for 15 s, 68 °C for 40 s, and then a final extension at 68 °C for 4 min. Then, the samples were indexed by a second PCR using the protocol in Feld et al. [26] and pooled in an equimolar amount before sequencing.

Raw sequences were analyzed through the Vsearch tool using default parameters [27]. Briefly, raw paired-end fastq sequences were merged, filtered by expected error 0.25, replicated, and sorted by size to create robust amplicons with minimum overlaps of 100 nucleotides and merge read sizes between 70 and 400 nucleotides (NCBI Bioproject PRJNA727863). Operational taxonomic unit (OTU) clustering was performed at 97% sequence identity, followed by quality filtering through de novo chimera removal using the UCHIME algorithm [28], considering in further analyses only groups with at least two sequences. Then, combined sequences were mapped to the list of OTUs with at least 97% identity, resulting in an OTU table with OTU sequences quantified in each treatment plot before and after treatment. The taxonomic annotation was performed using the SINTAX algorithm [29] by using k-mer similarity to identify the top taxonomy candidate, after which we retained the results where the species level had a score of at least 0.7 bootstrap confidence after comparing them with the SILVA database version 132 [30] and the UNITE database version 7.2 [31] as taxonomic references. The frequency tables for 16S and ITS were rarefied to 10,000 high-quality reads per table, and the abundance of each taxonomic group was calculated as a taxonomic percentage of the total amount of OTUs detected. The number of reads per sample and OTU is presented in Appendix A.

### 2.3. Edaphic Factors of the Plots Subjected to Different Irrigation Amounts and AMF Inoculation

After treatment application (30 July), samples of soils were collected at a depth of 20 cm. A cylindrical sample of 378.67 cm^3^ of bulk soil per sample was used to determine soil bulk density and soil water content (SWC). Soil samples were weighed and dried in an oven at 70 °C. Dried samples were used for determining the SWD and BD according to the following equations: (1)SWC ( gg)= weight of moist soil − weight of ‘oven−dried’ soilweight of oven dried soil
(2)Bulk density (gcm3)= weight of ‘oven−dried’ soilvolume of soil

Then, 20 g of soil samples diluted in deionized water (2:5, *w*/*v*) were used for determining the soil pH with an autotitrator (Omnis titrator, Methohm, Switzerland). Soil relative humidity (RH), temperature, and water evaporation from soils were measured in situ with a Soil Respiration Chamber (SRC-2) coupled to a Ciras-3 infrared gas analyzer system (PP Systems, Amesbury, MA, USA) at noon on 9 June. The SRC-2 chamber was stabilized for one minute, and then measurements were recorded.

### 2.4. Statistical Analysis

All the statistical analyses were carried out using R-Studio version 3.6.1 (RStudio: Integrated Development for R., Boston, MA, USA). Venn diagrams were generated using the package “VennDiagram” for R [32]. Then, OTUs present in less than two-thirds of the replicates were discarded to ensure reproducibility [27] before the statistical analyses. Alpha diversity indices (observed richness, Shannon, and inverse Simpson) were calculated and fit in linear mixed-effect models (LMEM) using the lmer function from the “*lme4*” package [33], with NT plots or different treatments based on irrigation amounts and AMF inoculation (FINI, FII, HINI, or HII) as fixed factors and replicates as random factors [34]. The significance of the models was tested with the “lmerTest” package [35]. Then, pairwise contrasts were conducted with function lsmeans from the “lsmeans” package [36] using the Kenward–Roger method and Tukey’s adjustment for *p*-values. The OTU table was used to perform the permutational analysis of variance (PERMANOVA) and the permutational analysis of multivariate homogeneity of groups dispersions (BETA-DISPER) using the functions Adonis and betadisper in the “vegan” package [37] with 999 permutations. Similarities or dissimilarities of the studied communities, those which resulted as significant from the PERMANOVA analysis, were plotted by non-metric multidimensional scaling analysis (NMDS) and principal coordinates analysis (PCoA). Differences in the relative abundance at the phylum and class levels between different treatments were assessed with LMEM, as described above. Cladograms of the taxonomy were drawn using the R package “Metacoder” [38]. The size of the nodes is proportional to the relative abundance of the taxa, while color represents changes in the different plots. Soil chemical and physical parameters were analyzed by LMEM with AMF inoculation (M), irrigation treatment (I), and their combination (M × I) as fixed factors and replicates as random factors; then, the significance of the model and the contrasts between the treated plots were tested, as described above. The influence of soil chemical and physical parameters on the microbial communities was explored by canonical correspondence analysis (CCA) using “vegan” [37] and “labdsv” [39] packages. Then, the significance of the CCA was evaluated with ANOVA. Finally, two networks were constructed based on time of sampling. Thus, a co-occurrence analysis of all treated plots/replicates was conducted before and after treatment (n = 16 per condition). Then, fungal and bacterial networks were separately constructed in each factor plot after treatment (FINI, FII, HINI, or HII). In every network, all replicates were considered (n = 8) to obtain a more accurate correlation between different OTUs. Co-occurrence networks were analyzed by using the “cooccur” package in R [40]. This method employs a probabilistic approach to determine which species pairs co-occur more (positive co-occurrence) or less often (negative co-occurrence) than expected. The analysis was conducted on the probabilistic model developed by Veech [41] based on pairwise comparisons of species presence/absence using, primarily, randomization (null model). Co-occurrence network visualization was conducted using the “VisNetwork” package [42] with the same software. 

## 3. Results

### 3.1. Richness and Diversity of Bacterial and Fungal Communities in Vineyard Plots before and after AMF Inoculation and Irrigation Systems Application

Three kingdoms were identified in the samples (Fungi, Archaea, and Bacteria) (Appendix A). Venn diagrams were generated before removing the less common species found in the trial (see the “Material and Methods” section) to assess the distinct and common bacterial and fungal species among different treatments (Appendix A). There were 97 bacterial and 273 fungal OTUs specific to the NT plots. On the other hand, there were 9 bacterial species specific to FINI and HII, whereas 19 bacterial OTUs were found in HINI as well as FII plots. We did not find specific fungal OTUs for plots after treatments. Thus, the number of bacterial species shared by all the plots was 46.4%, whereas, for fungi, they accounted for 21.4% of total observations.

The analysis of α-diversity indices in different plots showed differences according to the LMEM (Figure 1). Regarding bacterial communities, species richness was decreased in the FINI plots compared to the NT ones (Figure 1A). However, no differences were found between the different plots regarding the Shannon or inverse of Simpson indices (Figure 1C,E). Fungal communities were significantly affected by the treatments. Thus, the number of fungal species identified was significantly lower after treatment application, especially in the FII plots (Figure 1B). The Shannon index was decreased after treatment in the NI plots (Figure 1D). On the other hand, the inverse of Simpson index increased after treatment application in all the plots (Figure 1F).

In order to find possible effects of competition between bacteria and fungi, the correlation between α-diversity indices was conducted. Pearson’s correlation across all samples was negligible for Shannon and inverse of Simpson indices (r = 0.160, *p* = 0.39 and r = −0.095, *p* = 0.60, respectively) (Appendix A). Similarly, relationships of the diversity indices within each treatment/plot were insignificant, suggesting no competition between fungal and bacterial communities (Appendix A).

Correlations between microbiome structure and composition and the applied treatments were studied by computing between-sample diversity using the Bray–Curtis distance (Figure 2). For bacteria, principal coordinates analysis (PCoA) showed dissimilarities between samples from different treatments that clustered separately in three groups—NT, the full irrigated plots (FINI and FII), and the half irrigated (HINI and HII) plots, respectively (Figure 2A). The first two components explained 28.1% and 18.8% of the variation. Additionally, all the applied treatments (AMF inoculation and irrigation amounts), time of sampling, and their combination affected the bulk soil bacterial communities, as showed in Table 1. The ordination of the data in a non-metric multidimensional scaling (NMDS) indicated that all samples clustered closely, suggesting similar bacterial communities (Figure 2B). The application of treatments significantly affected beta-diversity; a PERMANOVA test showed a clear distinction (F = 4.879, *p* = 0.001). 

On the other hand, the PCoA for fungal communities highlighted changes in composition due to treatments, where NT samples were clearly separated from the treatment samples (Figure 2C). The first two components explained 55.8% of the total variance. Moreover, PERMANOVA tests comparing the different treatment combinations showed clear distinctions between them (Table 1). Likewise, the NMDS showed two clusters, with NT clearly separated from the samples after treatments (Figure 2D). The PERMANOVA test highlighted how treatments affected fungal beta-diversity (F = 7.615, *p* = 0.001). These results suggest that the main factor affecting the bacterial and fungal composition is the time of sampling and that bacterial composition is more responsive to different irrigation amounts than fungal composition.

### 3.2. Bacterial and Fungal Taxa Distribution in the Merlot Vineyard Bulk Soil Is Significantly Affected by AMF Inoculation, Irrigation Treatments, and Sampling Time

The taxonomic affiliations of the bacterial OTUs’ core showed that the vineyard bulk soil hosted 18 phyla, 38 classes, 71 orders, 130 families, and 298 genera (Appendix A). Different samples were dominated by *Proteobacteria* phylum that accounted for, on average, more than 35% in the bacterial communities. Other common phyla were *Actinobacteria* (ranged between 15.3% and 23.9%), *Verrucomicrobia* (between 8 and 12.2%), *Gemmatimonadetes* (between 4.8 and 9.8%), *Firmicutes* (between 2.3 and 7.7%), *Planctomycetes* (between 5.6 and 8.8%), and *Chloroflexi* (between 1.5 and 2.9%), as well as the phylum *Crenarchaeota* belonging to *Archaea* (ranged between 0 and 16%) (Figure 3A). Irrigation, AMF inoculation, and time affected the abundances at phylum levels, subject to the significance of the LMEM conducted (Appendix A). Thus, FI increased the proportion of *Proteobacteria* (*p* ≤ 0.0001), while reduced applied water with HI increased the abundances of *Actinobacteria* (*p* = 0.032), *Chloroflexi* (*p* = 0.019), and others (*p* ≤ 0.0001).

The 12 most abundant bacterial classes, which accounted for more than 90% of the relative abundance in all treatments, are presented in Figure 3C. Irrigation and AMF inoculation treatments affected the proportion of these bacterial classes, as highlighted with the significance of the LMEM (Appendix A). Therefore, we observed increased abundance of *Bacilli* and decreased abundance of *Gemmatimonadetes* compared to the abundances of NT (*p* ≤ 0.0001 for both classes). On the other hand, FI and/or AMF inoculation increased the abundances of *Alphaproteobacteria*, *Betaproteobacteria*, and *Gammaproteobacteria*. FI also decreased the abundance of *Thaumarchaeota* and other bacterial classes (*p* ≤ 0.0001 and *p* = 0.003, respectively, Appendix A). Finally, HINI decreased *Alphaproteobacteria* and *Betaproteobacteria* and increased *Gammaproteobacteria* and *Actinobacteria* classes. 

Regarding the bulk soil microbiome, different plots showed 4 phyla, 13 classes, 37 orders, 67 families, and 121 genera (Appendix A). The most abundant phyla we found were *Basidiomycota*, which, on average, accounted for 50% of the fungal communities, *Ascomycota* (ranged between 21.6% and 37.5%), and *Zygomycota* (ranged between 3.3% and 17.6%) (Figure 3B). Relative abundances of the fungal phyla were highly responsive to the treatments, as indicated by the LMEM (Appendix A). Thus, *Ascomycota* decreased in the HINI plots (*p* = 0.007), while *Basidiomycota* increased after treatment application in all plots (*p* ≤ 0.0001). The abundance of *Zygomycota* phylum increased with the combination of treatments, especially in FI and after inoculation with AMF (*p* ≤ 0.0001), whereas the abundances of other less abundant phyla decreased (Figure 3C, Appendix A).

Within fungal classes, the most abundant were *Agaricomycetes*, *Dothideomycetes*, *Eurotiomycetes*, *Leotiomycetes*, *Pezizomycetes*, *Saccharomycetes*, *Sordariomycetes*, and *Tremellomycetes*, which accounted for 50% of the fungal abundance in the NT plots and between 62% and 92% in the plots after treatment (Figure 3D). However, the distribution of fungal classes was strongly affected by treatments (Appendix A). Non-inoculated plots decreased the abundances of *Agaricomycetes* (*p* = 0.008), *Leotiomycetes* (*p* ≤ 0.0001), and *Saccharomycetes* (*p* = 0.001) especially in the FI plots. On the other hand, all the treatments increased the abundance of *Tremellomycetes* to the detriment of *Sordariomycetes* and *Dothideomycetes* classes. Finally, compared to NT, plots after treatment application decreased in the abundance of other fungal classes (*p* ≤ 0.0001; Figure 3D and Appendix A).

### 3.3. Bacterial and Fungal Clade Proportions in the Merlot Vineyard after Different Treatments of AMF Inoculation and/or Irrigation

Cladograms showed the analysis of the differences in the median proportions of each microbiome family due to treatments (Appendix A). Within the Proteobacteria phylum (Appendix A), HI increased the proportion of Oceanospirillaceae and Cellvibrionaceae families. FII increased the Marinicellaceae, Anaeromyxobacteriaceae, and Rickettsiaceae families and decreased Neisseriaceae. The main changes in Actinobacteria (Appendix A) were increments of the proportions of Streptoporangiales and Cryptosporangiaceae due to HI and the enhancement of Micromonosporaceae, Acidimicrobiaceae, and Actinospicaceae in FII, HINI, and HII plots. Firmicutes phylum proportions were highly affected by treatments (Appendix A). Thus, HI increased the proportion of members of the family XVII and Erysipelotrichaceae. FII increased Syntrophomonadaceae and Gracilibacteraceae proportions. Finally, compared to FINI plots, FII, HII, and HINI enhanced Halanaerobiaceae. Within the other less abundant phyla (Appendix A), the main differences were due to HI, which increased families from Chlamydiales and Spirochaetia clades and enhanced the proportions of families belonging to the Archaea kingdom. On the other hand, FII increased the proportion of the Deinococcaceae family.

Regarding the effect of different treatments on the proportion of fungal families, members of the Ascomycota phylum were highly affected by treatments (Appendix A). FII increased the proportion of Glomerellaceae and Togniniaceae, HII increased Sporormiaceae, Tubeufiaceae, and decreased Eremomycetaceae, whereas FI led to increased proportions of Leptosphaeriaceae. A Basidiomycota phylum cladogram (Appendix A) showed that different irrigation treatments affected the Malasseziaceae (increased in HI) and Sparassidaceae (increased in FI) families. On the other hand, AMF inoculation increased the proportions of Hydnodontaceae under FI conditions and Psathyrellaceae and Cortinariaceae under HI conditions. Zygomycota families were not highly affected by the treatments (Appendix A).

### 3.4. Edaphic Factors Barely Affected in Bacterial and Fungal Distribution

The SWC decreased in the HI treatment that accounted for a decreased soil evaporation, especially in HII plots, although no interactive effect between the factors was evident (Table 2). The edaphic factors of different plots slightly influenced the bacterial and fungal communities, as showed in the canonical correspondence analysis (CCA) (Figure 4, Appendix A). However, Figure 4A shows that soil temperature and bulk density, soil evaporation, and soil water content are correlated with bacterial beta-diversity and explains the separation between HI and FI treatments. This separation is related to the abundances of *Nitrososphaera* sp., *Jatrophihabitans* sp., *Actinophytocola* sp., *Pseudonocardia* sp., *Geodermatophilus* sp., *Actinomycetospora* sp., and *Rugosimonospora acidiphila* or *Micromonospora hermanusense* species, as shown in the correlation with CCA1 (Appendix A).

On the other hand, the treatments applied did not differ in their fungal composition driven by edaphic factors, and all the treatments clustered together (Figure 4B). However, soil evaporation and the SWC were negatively correlated with fungal composition, which could be related to the abundance of *Penicillium*, *Aspergillus*, *Cryptococcus*, and *Cladorrhinum* members (Appendix A).

The analyses of the co-occurrence of bacterial and fungal networks in the bulk soil of the Merlot vineyard showed different connectivity patterns influenced by time (before and after treatment application; Figure 5, Table 3) and by the different treatments (FINI, FII, HINI, and/or HII; Figure 6, Table 4). 

Before treatment, just 973 (0.7%) of the 141,796 pairwise comparisons yielded statistically significant co-occurrence, comprising 615 positive and 358 negative associations (Table 3). Similarly, after treatment application, from the 135,356 pairwise comparisons, only 1049 were statistically significant, with 795 positive and 254 negative co-occurrences, respectively. Thus, although the total number of co-occurrences did not differ between samples before and after treatment application, the latter showed more positive and less negative associations compared to the pretreatment samples (Table 3, Figure 5). This shift was likely influenced by the enhancement of positive associations between bacteria species and the diminution of negative fungal associations in soil samples after treatment application (Table 3).

We also conducted co-occurrence analyses to assess the effect of AMF inoculation and irrigation treatments (Figure 6, Table 4). In the FI plots, 229 co-occurrences were found to be statistically significant; the majority of them were bacterial associations, bacteria–fungi associations, and, less frequently, associations between fungal species. Around 57% (130) of the co-occurrences in the FI plots were positive, while 43% were negative (99, Table 4). 

Figure 6A shows that significant negative co-occurrences happened between bacterial and fungal species separately, for instance, *Chloroflexi* and *Acidobacteria* with *Proteobacteria* or connections between *Ascomycota* species. The positive connectivity found in the FI plots was mainly explained by associations between bacteria species, such as the links of *Bacteroidetes*, *Firmicutes*, *Proteobacteria*, and *Actinobacteria* species. On the other hand, HI plots had 300 significant associations, and 77% (231) of the connections between microbial species were positive, while 23% (69) were negative. In these plots, half of the connections were identified as fungi–bacteria associations. Thus, great connectivity between species of *Acidobacteria* or *Proteobacteria* with *Ascomycota* and *Basidiomycota* members was found (Figure 6B).

Regarding the NI plots, 271 significant associations were found, with 176 being positive and 95 being negative (Table 4). The majority of the associations were between bacteria species (about 45%, Table 4) or between bacteria and fungi (39%). Negative associations were more frequently between species belonging to *Acidobacteria*, *Proteobacteria*, *Bacteroidetes*, or *Chloroflexi* phyla, whereas positive associations were found between *Actinobacteria*, *Verrucomicrobia*, *Proteobacteria*, and other bacteria phyla (Figure 6C). Moreover, a great degree of connectivity (either positive or negative) between *Acidobacteria* and *Proteobacteria* with *Ascomycota*, *Basidiomycota*, and *Zygomycota* species was highlighted. Finally, inoculated plots showed 263 significant associations, with 75% being positive co-occurrences (196) and 25% being negative (67, Table 4). Again, the majority of connections were within bacteria species or between bacteria and fungi species. After inoculation, associations between fungi and bacteria were found, such as *Proteobacteria* with *Ascomycota* or *Verrucomicrobia* with *Basidiomycota*. Contrarily, the connectivity between fungal species was very low. The increase in the negative connectivity network of *Acidobacteria* species, especially with *Verrucomicrobia* and bacteria belonging to other clades, and *Actinobacteria* phyla with fungal and bacterial species is noteworthy (Figure 6D).

## 4. Discussion

### 4.1. Differential Responses of Bacterial and Fungi Alpha and Beta Diversities to AMF Inoculation and Irrigation Treatments

This study provides evidence that cultural practices affect the abundance and relationships between bacterial and fungal communities differently. The analysis of vineyard microbial communities has recently been studied, but the studies were limited to the examination of different soils characterized by geography [14,43,44], vineyard management [14,45,46], rootstocks [12,47,48], and rotundone zones (spatial variability of soil water supply affecting secondary metabolites) [49]. Our aim, however, is to expand the understanding of how the management practices (AMF inoculation and irrigation amount) affect the microbiome associated with replanted grapevines by studying them in a short period. 

The bacterial core, meaning the number of shared species between different treatments, was higher than the fungal core. According to Coller et al. [14], the majority of bacterial OTUs were present in a small number of samples, whereas a small number of OTUs were shared by all samples, showing a high degree of variability across samples. In this study, fungal and bacterial communities responded differently to the irrigation treatments and the AMF inoculation, according to the alpha diversity indices presented, where bacteria species richness was not affected by the treatments in contrast to a previous study [14]. Willing et al. [50] reported that increasing water availability across the coastal redwood range decreased bacterial species richness, estimated as the Shannon index, suggesting that the turnover in bacterial communities was most likely to be driven by species loss, with increasing water availability instead of species replacement. This was noticed in this study, where the Shannon index increased in half irrigated plots. Under our experimental conditions, bacterial alpha diversity was higher than fungal alpha diversity, in accordance with Liang et al. [51], who suggested that bacteria probably play roles that are more pivotal than fungi in vineyard soils.

Fungal richness decreased after treatment application, especially in the plots subjected to full irrigation. Conversely, Alonso et al. [52] found that the fungal population appeared to be more stable (compared to bacterial communities) in terms of alpha diversity, while Coller et al. [14] did not report a significant effect of different land use on core soil mycobiome. However, our results are corroborated by Zhang et al. [53], who reported that after 36 years of irrigation or straw cover management of a wheat production area in China, fungal alpha diversity indices were decreased with irrigation compared to straw cover treatment, whereas no differences were found in bacterial alpha diversity indices. It is known that the composition of fungal communities in soils is very responsive to plant root–exudate composition, which would vary with plant phenology, especially during flowering and ripening [54] and/or responding to environmental conditions [55]. 

The PCoA and NMDS analyses suggest a weak effect of AMF inoculation, whereas the irrigation applied and, especially, the time course played major roles in selecting the bacterial component of the microbiome in vineyard bulk soil. Therefore, the alteration of soil moisture through irrigation practices may influence the abundance, structure, and function of soil microorganisms, which likely modified the effect of the irrigation program on vine performance [56]. It is known that irrigation management affects carbon partitioning in grapevine organs, including roots [57], and thus might also increase the content of simple sugars in root exudates, which, as a non-specific, easily accessible resource, stimulates the entire active microbial community during recovery from drought [58].

The PCoA and NMDS of the Bray–Curtis distances for fungal communities showed that the date of sampling was the main source of beta-diversity. Overall, bacterial communities seemed to be more responsive to treatments and time course than fungal communities, corroborating previous studies in microbiome analysis [9]. Accordingly, several studies documented the temporal heterogeneity based on the inter-annual variability in vineyard soils [43,48,59,60], but few of them studied this effect within the same season [43,61]. Therefore, microbiome studies should consider the high degree of temporal variability in the experimental design because sampling the same plot at different times can give different results due to the variability of its own microbial community over time [62].

### 4.2. Bacterial and Fungal Composition after AMF Inoculation and Irrigation Treatment Application during the Season

*Proteobacteria* and *Actinobacteria* phyla were predominant in the vineyard soil, covering about 50% of the bacterial relative abundances in accordance with previous research [43,46,47,49,51,61] suggesting that the bacterial microbiome in bulk soil is partially conserved. Furthermore, recent research with a proteomic approach suggested that *Proteobacteria*, *Actinobacteria*, and *Firmicutes* were the most active phyla in vineyards in protein expression and were mainly involved in phosphorus and nitrogen rhizosphere metabolism [63] and in the carbon biochemical cycle and their production of secondary metabolites [64]. Under our experimental conditions, *Actinobacteria* and *Chloroflexi* phylum increased in the HI plots. Accordingly, it was previously reported by Moreno-Espíndola et al. [65] in an organic Milpa farm with different moisture; thus, these authors suggested that phylotypes belonging to *Actinobacteria* and *Chloroflexi* were enriched in dry conditions because their thick cell walls, filamentous growth, and spore formation favored them under dry conditions [66]. Furthermore, it was recently reviewed by De Vries et al. [67] that drought increases the secondary metabolites in root exudates, stimulating specific microbial groups, which potentially increase N availability for the plant, and the exudation of organic acids to mobilize inorganic P, which has the potential to prime the decomposition of soil organic C through the stimulation of the decomposer’s growth.

At the class level, *Alphaproteobacteria* was the dominant class, with frequencies about 20% in all the soil samples, followed by *Actinobacteria* (~10%), *Spartobacteria* (~9%), *Gammaproteobacteria* (~9%), and *Thaumarchaecota* (~8%). Several endophytic microorganisms belonging to *Actinobacteria* have been reported to control grapevine pathologies, given their in vitro antifungal activity against the main fungal pathogens affecting young grapevines in nurseries, thus decreasing their mortality and infection rates [68,69]. Under our experimental conditions, AMF and/or FI treatments increased the proportions of *Alphaproteobacteria*, *Betaproteobacteria*, and *Gammaproteobacteria*. Glomalin is secreted by the AMF hyphae to stabilize soil aggregates, and it is responsive to environmental factors, which may also have important implications and connections to carbon sequestration, improving soil health and the biodiversity of soil microorganisms [70].

At the phylum level, *Ascomycota* and *Basidiomycota* were the most abundant phyla detected in all samples, accounting for about 50% before treatment applications and for almost 80% of the relative abundance after treatment application. Previous studies also agreed on the most common fungal phyla detected in vineyard soils [6,14,43,44,51,71]. After AMF inoculation, an increase in *Ascomycota* and *Zygomycota* relative abundances was recorded. *Ascomycota* phylum participates in the decomposition of organic matter [72] and is known to respond to small-scale changes in soil chemistry, water, and resource concentrations rather than geomorphic land system classifiers [73]. On the other hand, members of the *Zygomycota* phylum have been pointed out as suppressors of soil–plant pathogens [74].

### 4.3. Irrigation Treatments and AMF Inoculation Shifted Microbial Communities but Not through Changes in Soil Edaphic Factors

AMF inoculation and irrigation treatments slightly affected the physicochemical properties of soils, where only soil water evaporation and soil water content were enhanced in the FI plots. These results agree with previous studies that did not find changes in pH and soil moisture in spite of the intra-vineyard variated zones and soil depths [51]. Soil physicochemical properties and moisture content have been identified as major factors shaping the spatial scaling of the grapevine rhizosphere microbiome in many previous studies [4,11,56]. Our results partially support this, according to the two CCAs conducted. We found that edaphic factors were barely correlated with bacterial and fungal community dissimilarities. 

The main changes in *Proteobacteria* phylum composition were driven by irrigation treatments. Thus, HI soils increased the members of *Oceanospirillaceae* and *Cellvibrionaceae* families. This is in agreement with a recent study that proposed that new species of the *Oceanospirillaceae* family may help rice to overcome saline stress and promote plant growth by increasing ACC activity [75]. In addition, the increment of *Cellvibrionaceae* may be related to potential saline–alkaline stress, as reported in a previous study dealing with Cadmium-contaminated soils [76]. The HI treatment enhanced the presence of genera belonging to *Actinobacteria*, which are known to be endophyte-related, to impair the decline of young grapevines caused by the fungal trunk pathogen infection through the root system in nurseries [69]. This enhancement was mainly due to the increased presence of genus *Fodinicola*, which was recently suggested as a promising source of novel secondary metabolites for enhancing plant growth [77]. In addition, AMF and HI enhanced the presence of *Micromonosporaceae*, related to enhancing the health in N-fixing plants and the growth of several host plants under controlled conditions [78]. In these treatments, the presence of members belonging to the *Actinospicaceae* family and Candidate division TM7 increased. *Actinospicaceae* improved antifungal activity, siderophore production, and phosphate solubilization activity [79], while Candidate division TM7 is a key biomarker of the resistance against wilt disease in tobacco [80] and after *Verticillium* inoculation in olive [81]. Overall, these changes may lead to better soil quality.

With regard to the fungi, *Ascomycota* composition was highly affected by AMF inoculation and irrigation treatments. Thus, AMF inoculation under FI conditions increased the proportion of plant pathogens such as *Phaeoacremonium* (*Togniciaceae* family), which has been related with Esca disease in grapevines [82] and *Glomerella tucumanensis* (*Glomerellaceae* family), responsible for the red root rot disease in sugarcane, affecting sugar production and productivity [83]. However, under HI conditions, AMF inoculation increased the proportion of *Preussia* (*Sporormiaceae* family), an endophytic fungus isolated from *Vitis labrusca* L. leaves without in vitro activity against the pathogen *Fusarium oxysporum* [84]. The FI treatment also increased the proportion of *Coniothyrium* (*Leptosphaeriaceae*), which was suggested as a potential biocontrol of *Sclerotinia* lettuce drop [85]. 

Within Basidiomycota, AMF inoculation increased the presence of the *Psathyrellaceae* family, whose members are known to be endophytes that may contribute to grapevine growth and productivity [86] and are very abundant in organic farmlands [87]. Inoculated soils also increased the presence of *Hydonodontaceae*, which has been proposed as a potential bioindicator, given the negative relationship with ginseng (*Panax notoginseng*) mortality in a continuous cropping system [88].

### 4.4. Time Course and Management Practices Affect Bacterial and Fungal Co-Occurrence Networks

Network analysis of microbial communities is a useful tool that allows the assessment of community structure and potential interactions between its members [89]. Under our experimental conditions, the co-occurrence analyses revealed the majority of ‘random’ associations between species. Two species can be classified as a random association when the statistical power is not sufficient, although the difference between observed and expected co-occurrence is substantial, or when the association between two species is truly random, with similar observed and expected co-occurrences [41]. Further studies with an increased number of sampled sites could distinguish the associations classified as random. Comparing the networks before and after treatment, we showed evidence that the number of positive associations between microbes increased. In other words, the observed co-occurrence of two species was greater than the expected co-occurrence, and species tended to occur together at more locations than expected when they were randomly distributed, relative to the other species. Contrarily, negative associations were decreased, namely, two species co-occurred at fewer locations than expected, given that they were randomly distributed. This led to a more complex soil microbial community network along the course of the season, which has been suggested to benefit plants [90]. Indeed, Layeghifard et al. [91] suggested that the functional capacity of the microbiome is not equal to the sum of its individual components, given the interaction between microbial species and the formation of complex networks that significantly influence ecological processes and host adaptations. Our results corroborated this hypothesis; although species richness decreased along time, especially in the fungal community, the number of positive associations increased after treatments accounted for 24% of connections between bacteria species, 62% between fungal species, and 15% of fungi–bacteria associations. It is worth mentioning that co-occurrence network visualization showed the correlative relationships between taxa, including the true ecological interactions (i.e., mutualism) and also nonrandom processes (i.e., niche-overlap), and therefore, they do not necessarily reflect direct interactions between species [16]. Nevertheless, they were still convenient for exploring abundance patterns in complex microbial communities and evaluating the effects of different types of management. 

The comparison of treatments showed different patterns of connectivity within after-treatment samples. Thus, compared to FI, HI increased the connectivity between species, increasing the number of positive co-occurrences and promoting the associations between fungal species and fungal and bacterial species. Accordingly, de Vries et al. [17] found that soil bacterial networks were less stable under drought than fungal networks. Thus, these authors reported that drought stress promoted destabilizing properties of bacterial co-occurrence networks via changes in vegetation composition and resultant reductions in soil moisture.

On the other hand, the comparison of I and NI plots showed a similar degree of connectivity. However, the inoculation with AMF promoted the positive associations, to the detriment of the negative associations, between species. Previous studies reported that the inoculation of plants with plant-growth-promoting bacteria led to more complex and compact associations in their associated microbiome. Accordingly, the inoculation of *Camelia sinensis* with a microbial consortium strengthened the connection between the bacteria, indicating greater stability of the community compared to the control and promotion of cooperation after inoculation [22]. In the soybean rhizosphere, inoculation with *Rhizobium*, a natural N-fixing bacteria symbiont of leguminous plants, led to an increased number of connections between fungal species [23]. In contrast, in the rhizosphere of AMF-inoculated onions, AMF co-occurred with several indigenous bacteria and fungi, suggesting that AMF may recruit specific taxa to confer better plant performance [21]. Similar to these works studying the effect of symbionts on the rhizosphere microbiome, our study corroborated that the effect is conserved in the bulk soil. Furthermore, the observed changes in network connectivity were likely explained by the modification of plant signaling molecules, hormones, and exudate composition that take place with mycorrhizal symbiosis and are known to modify soil characteristics [18]. Currently, little experimental evidence is available, and our work gives some insights on the effects that inoculation with AMF and irrigation management may have on microbial community structure and connectivity. Nevertheless, the findings presented here require further studies with, for instance, a greater number of locations with different climatic conditions and vineyard ages, given the variability in rhizosphere microbiomes between locations [14,43,44] and young and mature grapevines [43,68,71].

## 5. Conclusions

We conducted this experiment to assess whether the vineyard soil microbiome was altered by different management practices such as the use of biostimulants (AMF inoculation vs. non-inoculated) and/or irrigation management (fully irrigated vs. half irrigated). Our results indicate that bacterial and fungal communities in vineyard soils are shaped by both time course and soil management (i.e., the use of biostimulants or irrigation), and, more importantly, there was an interactive effect observed among these factors. Alpha diversity was more responsive to treatments in the fungal communities, whereas changes in beta diversity were mainly recorded in the bacterial communities. Microbial network analyses suggested that bacterial associations were weaker than the fungal ones under half irrigation and that the inoculation with AMF led to the increase of positive associations between vineyard-soil-living microbes. Altogether, our results highlight the necessity of more studies on the effect that management practices, especially the addition of AMF to cropping systems, may have on the soil microbiota in order to strengthen beneficial microbial networks and, consequently, achieve better soil quality, which will improve crop performance.

## Figures and Tables

**Figure 1 microorganisms-09-01273-f001:**
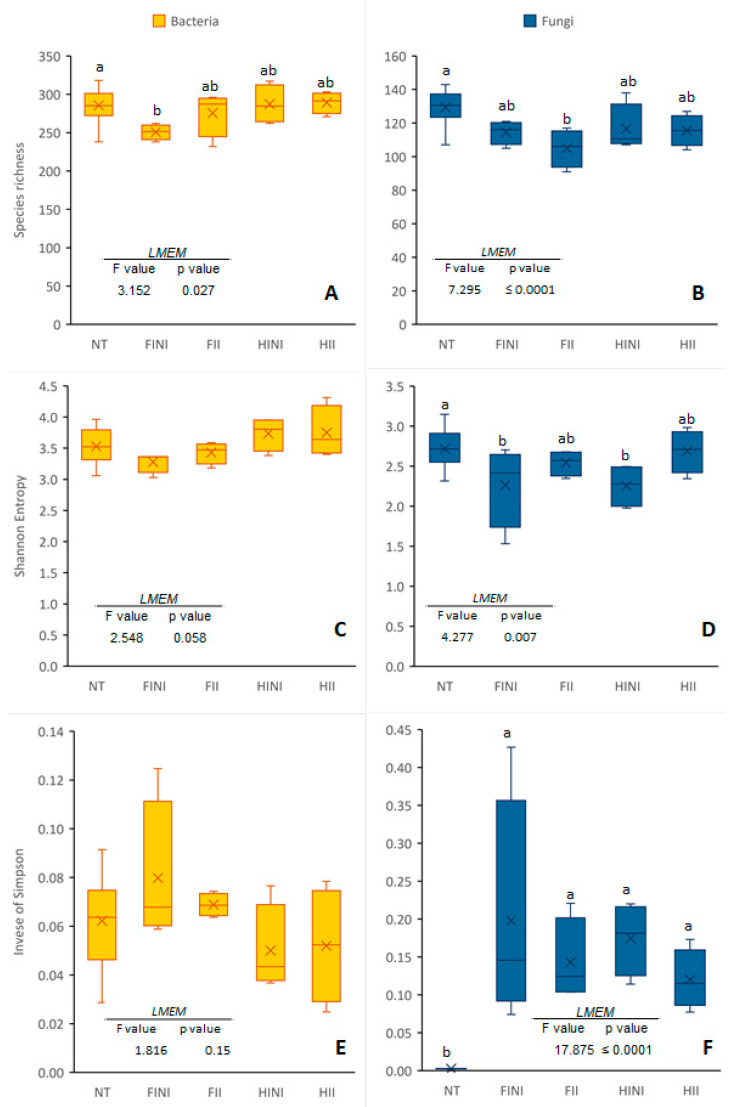
Species richness (**A**,**B**), Shannon diversity (**C**,**D**), and inverse of Simpson (**E**,**F**) indices of bacterial and fungal communities found in plots before (NT) and after treatments, different irrigation amounts (FI, fully irrigated, or HI, half irrigated), and AMF inoculation (I, inoculated, or NI, non-inoculated). Values represent means ± SE (n = 4–16), separated by the Kenward–Roger method and Tukey’s *p*-value adjustment (*p* ≤ 0.05). Different letters indicate significant differences, as affected by NT or treatment application (FINI, FII, HINI, or HII), according to the linear mixed-effect model.

**Figure 2 microorganisms-09-01273-f002:**
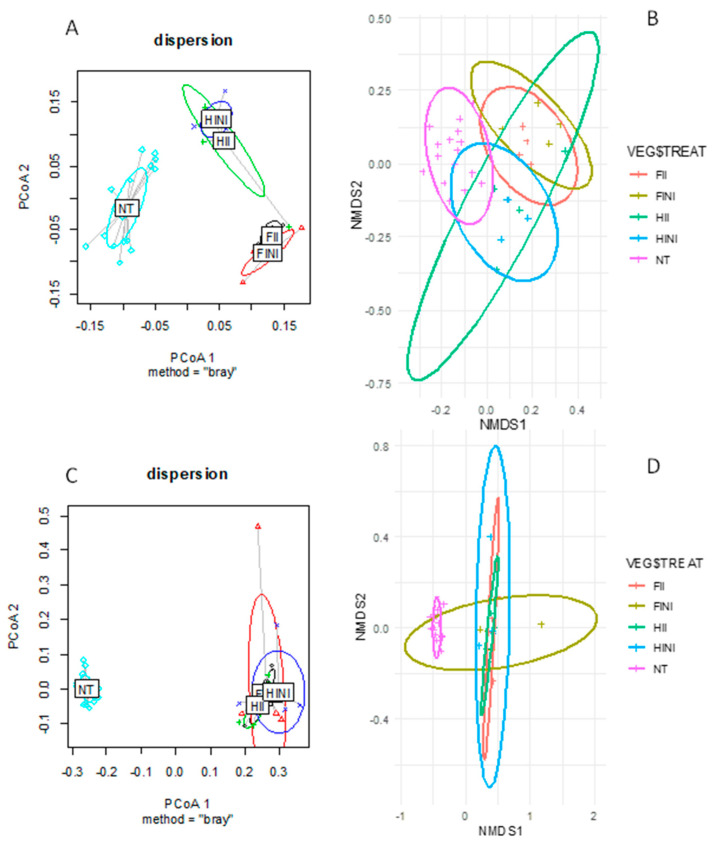
Principal coordinates analysis (PCoA) of the Bray-Curtis distance matrix for bacteria (**A**) and fungi (**C**) communities and the non-metric multidimensional scaling (NMDS) plot of Bray-Curtis dissimilarities for bacteria (**B**) and fungi (**D**) communities from vineyard soils subjected to different irrigation amounts (FI, fully irrigated, or HI, half irrigated), AMF inoculation (I, inoculated, or NI, non-inoculated) and their combinations.

**Figure 3 microorganisms-09-01273-f003:**
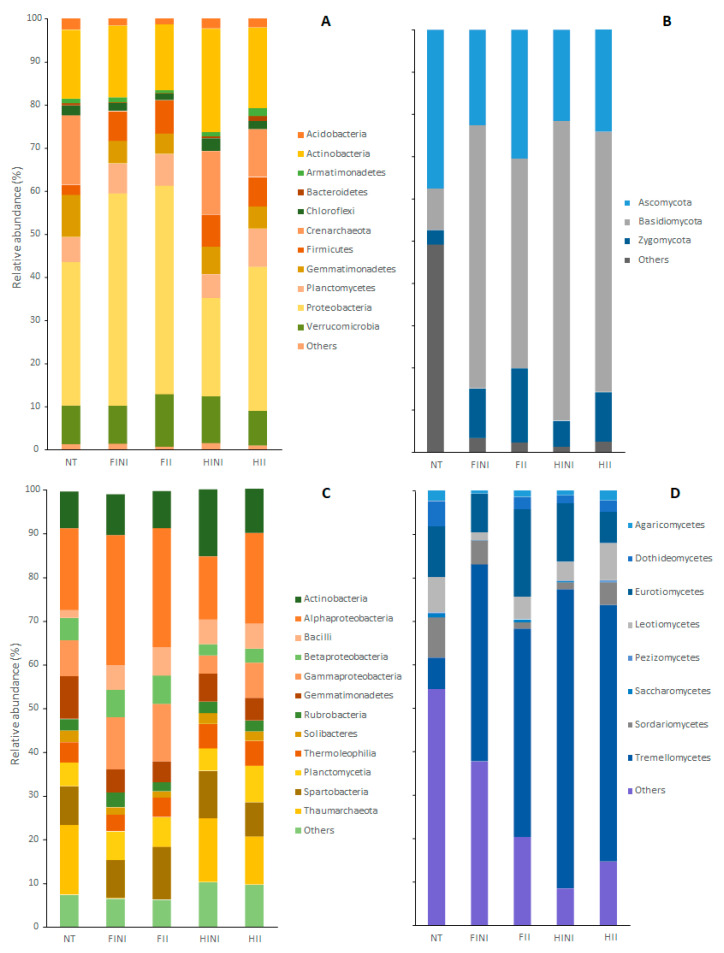
Bacterial and fungal community composition found in plots before (NT) and after treatments, different irrigation amounts (FI, full irrigated, or HI, half irrigated), AMF inoculation (I, inoculated, or NI, non-inoculated), and their combination at phylum (**A**,**B**) and class (**C**,**D**) levels.

**Figure 4 microorganisms-09-01273-f004:**
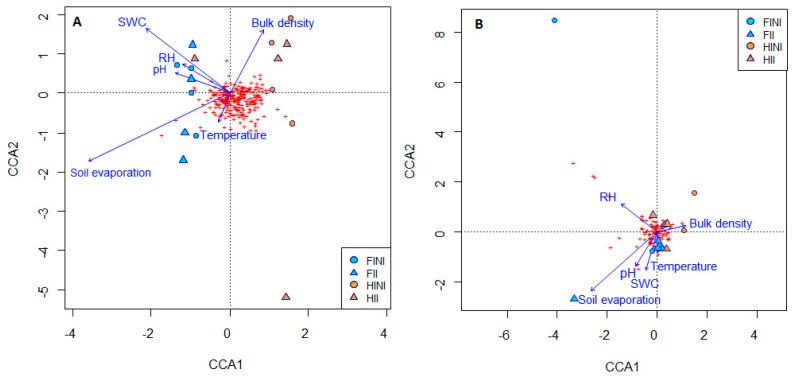
Canonical correspondence analysis (CCA) showing the correlation between soil edaphic factors and bacterial (**A**) and fungal (**B**) communities from vineyard soils subjected to different irrigation amounts (FI, Full Irrigated or HI, Half Irrigated), AMF inoculation (I, inoculated; or NI, non-inoculated) and their combinations.

**Figure 5 microorganisms-09-01273-f005:**
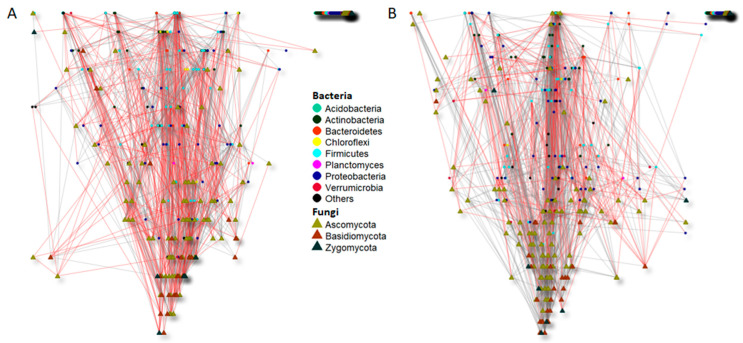
Significant co-occurrence networks of bulk soil bacterial and fungal communities of the Merlot vineyard before treatment application (13 March) (**A**) and after treatment application (16 June) (**B**). The nodes of each network are colored and shaped according to phylum affiliation. The edges connecting the nodes show negative (indicated with red lines) and positive (indicated with grey lines) associations between each species.

**Figure 6 microorganisms-09-01273-f006:**
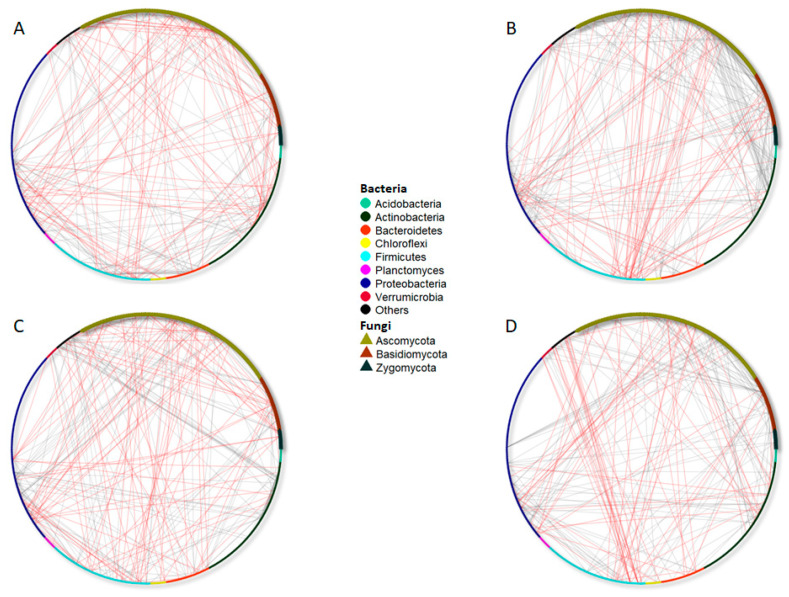
Significant co-occurrence networks of bulk soil bacterial and fungal communities living in the fully irrigated (FI, (**A**), half irrigated (HI, (**B**), non-inoculated (NI) (**C**), or inoculated with AMF (I) (**D**) plots. The nodes of each network are colored and shaped according to phylum affiliation. The edges connecting the nodes show the negative (indicated with red lines) and positive (indicated with grey lines) associations between each species.

**Table 1 microorganisms-09-01273-t001:** F and *p* values of PERMANOVAs comparing different irrigation systems (I), AMF inoculation (M), and time course (T) of Merlot vineyard microbial beta diversity.

		Bray–Curtis			
		Bacteria		Fungi	
Treatment	Comparison	F	*p*-Value	F	*p*-Value
Irrigation (I)	FI vs. HI	4.727	0.001	6.480	0.001
AMF inoculation (M)	NI vs. I	3.528	0.002	5.446	0.005
Time course (T)	T0 vs. T1	9.688	0.001	26.762	0.001
I × M	FINI vs. FII vs. HINI vs. HII	3.424	0.001	4.598	0.001
I × T	FI_T0 vs. FI_T1 vs. HI_T0 vs. HI_T1	8.788	0.001	14.134	0.001
M × T	I_T0 vs. I_T1 vs. NI_T0 vs. NI_T1	5.312	0.001	14.092	0.001
I × M × T	FINI_T0 vs. FINI_T1 vs. HINI_T0 vs. HINI_T1 vs. FII_T0 vs. FII_T1 vs. HII_T0 vs. HII_T1	4.879	0.001	7.615	0.001

**Table 2 microorganisms-09-01273-t002:** Edaphic factors of Merlot vineyard soil subjected to different irrigation amounts (FI, Full Irrigated or HI, Half Irrigated), and AMF inoculation (I, inoculated; or NI, non-inoculated).

	Soil pH	Relative Humidity (RH)	Soil Evaporation (mmol·m^−^^2^ s^−^^1^)	Soil Temperature (°C)	SWC (g·g^−^^1^)	Bulk Density (g·cm^−^^3^)
FI	5.90 ± 0.10	17.4 ± 1.2	0.41 ± 0.03 ^a^	39.1 ± 1.0	0.06 ± 0.01 ^a^	1.11 ± 0.02
HI	5.74 ± 0.09	16.1 ± 0.5	0.24 ± 0.02 ^b^	39.2 ± 0.5	0.05 ± 0.01 ^b^	1.12 ± 0.01
NI	5.74 ± 0.05	16.2 ± 0.9	0.29 ± 0.03 ^b^	39.7 ± 0.5	0.05 ± 0.01	1.11 ± 0.02
I	5.89 ± 0.12	17.3 ± 1.0	0.36 ± 0.04 ^a^	38.6 ± 0.9	0.06 ± 0.01	1.11 ± 0.02
*LMEM*						
Irrigation (I)	0.194	0.232	0.0001	0.862	0.029	0.591
AMF inoculation (M)	0.155	0.267	0.017	0.199	0.094	0.935
I × M	0.267	0.780	0.413	0.645	0.331	0.264

Values represent means ± SE (n = 8) separated by Kenward-Roger method and Tukey’s *p*-value adjustment (*p* ≤ 0.05). Different letters indicate significant differences as affected by treatment application (FI, HI, NI, or I) according to the main factors in the linear mixed-effect model.

**Table 3 microorganisms-09-01273-t003:** Degree of connection for bacterial and fungal communities found in Merlot vineyard plots before (not treated, NT) and after treatments, different irrigation amounts (FI, fully irrigated, or HI, half irrigated), and AMF inoculation (I, inoculated, or NI, non-inoculated).

		Not Treated (NT)	After Treatment
Positive connections			
	Total	615	795
	Bac–Bac	281	350
	Fun–Fun	126	205
	Bac–Fun	208	240
Negative connections			
	Total	358	254
	Bac–Bac	93	85
	Fun–Fun	85	18
	Bac–Fun	180	151
Total connections			
	Total	973	1049
	Bac–Bac	374	435
	Fun–Fun	211	223
	Bac–Fun	388	391
	Total analyzed pairs	141,796	135,356
	Percentage of non-random	0.7	0.8

**Table 4 microorganisms-09-01273-t004:** Degree of connection for bacterial and fungal communities in Merlot vineyard soils subjected to different irrigation amounts (FI, fully irrigated, or HI, half irrigated) or AMF inoculation (I, inoculated, or NI, non-inoculated).

		FI	HI	NI	I
Positive connections					
	Total	130	231	176	196
	Bac–Bac	72	52	97	66
	Fun–Fun	24	81	26	62
	Bac–Fun	34	98	53	68
Negative connections					
	Total	99	69	95	67
	Bac–Bac	30	24	25	28
	Fun–Fun	21	0	17	0
	Bac–Fun	48	45	53	39
Total connections					
	Total	229	300	271	263
	Bac–Bac	102	76	122	94
	Fun–Fun	45	81	43	62
	Bac–Fun	82	143	106	107
	Total analyzed pairs	120,299	128,975	128,097	126,817
	Percentage of non-random	0.2	0.8	0.2	0.2

## Data Availability

The raw fastq sequences are available at NCBI Bioproject PRJNA727863, and the identification and number of reads per sample are available in Appendix A. The dataset supporting the conclusions of this article is included within the article (Appendix A).

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
