# Peer review of "Inoculation with Mycorrhizal Fungi and Irrigation Management Shape the Bacterial and Fungal Communities and Networks in Vineyard Soils"

_microorganisms, 2021, doi:10.3390/microorganisms9061273_

Round 1

Reviewer 1 Report

The effort made by the authors is very valuable, as information about how AMF and irrigation effect the plant associated microbionta are lacking.

The manuscript fits within the scope of the journal. The title is clear and it is adequate to the content of the article. The conclusions are supported by the content. I have only two minor observations and a recommendation:

In the hypothesis it is mentioned that irrigation management and/or AMF inoculation may affect soil chemical properties. The article does not deal with how irrigation or AMF inoculation effect soil chemical properties. Only soil reaction (pH) was measured. Soil water content, bulk density, temperature, water evaporation are edaphic parameters. So hypothesis should be reformulated. 

Although in the "site description and experimental design" section it would be necessary a general characterization of the soil (chemical characteristics and texture).

line 93 - please change "Convulvulus" with "Convolvulus

line 520 - please change "Thaumarchacota" with "Thaumarchaeota"

Author Response

In the hypothesis it is mentioned that irrigation management and/or AMF inoculation may affect soil chemical properties. The article does not deal with how irrigation or AMF inoculation effect soil chemical properties. Only soil reaction (pH) was measured. Soil water content, bulk density, temperature, water evaporation are edaphic parameters. So hypothesis should be reformulated. 

We have revised as requested by the reviewer. 

Although in the "site description and experimental design" section it would be necessary a general characterization of the soil (chemical characteristics and texture).

We have revised as requested by the reviewer in this new version 

line 93 - please change "Convulvulus" with "Convolvulus" .  We have made the change as requested. 

line 520 - please change "Thaumarchacota" with "Thaumarchaeota" We have made the change as requested. 

Reviewer 2 Report

Here is the review of the manuscript entitled "Inoculation with mycorrhizal fungi and irrigation management shaped the bacterial and fungal communities and networks in vineyard soils" written by Nazareth Torres and co-authors.

The aim of the study was to assess whether the vineyard soil microbiome was changed after the use of biostimulants (made from arbuscular mycorrhizal fungi) in combination with different irrigation management regimes (full irrigation vs. half irrigation). Bacterial and fungal soil communities were influenced by different soil management regimes (biostimulants application and irrigation). Fungal communities responded more intensively to soil treatments (measured by alpha diversity), while changes in beta diversity were more expressed in bacterial communities. Soil factors mostly did not influence bacterial and fungal communities. Microbial network analyses indicated that bacterial associations are weaker than the fungal associations under half irrigation management regime. The authors conclude that more research is needed to assess the effect of management practices in the field, especially regarding the addition of AM fungi to agricultural systems to improve soil and consequently crop quality.

The study is very interesting, well prepared and conducted. English is very good. Field and lab methods are suitable for this kind of research. The study has no major flaws. Methods section, esp. soil sampling part needs to be clarified. Other than that, there are only a few typographical errors in the manuscript (please see pdf file attached).

I recommend the publication of the paper after minor revision.

Best,

Reviewer

Author Response

We have responded to the changes as requested by Reviewer #2 in the new version of the manuscript. 

Methods section, esp. soil sampling part needs to be clarified.

We clarified this soil sampling section. 

 Other than that, there are only a few typographical errors in the manuscript (please see pdf file attached).

The typographical errors were fixed in the new version.